# T Lymphoblastic Lymphoma Hiding in Mature Plasmacytoid Dendritic Cell Proliferation: A Case Report and Literature Review

**DOI:** 10.3390/diagnostics13203248

**Published:** 2023-10-19

**Authors:** Cong Deng, Beibei Gao, Tianli Wang, Xiaona Chang, Guixiang Xiao, Qin Xia, Huaxiong Pan, Xiu Nie

**Affiliations:** Department of Pathology, Union Hospital, Tongji Medical College, Huazhong University of Science and Technology, Wuhan 430022, China; claredeng1104@163.com (C.D.); beibei_gao0119@163.com (B.G.); wangtianli1999@163.com (T.W.); pathologycxn@163.com (X.C.); xiaoguixiang80@163.com (G.X.); xiaqinblk@163.com (Q.X.)

**Keywords:** mature plasmacytoid dendritic cell proliferation, T lymphoblastic lymphoma, myeloid neoplasms, case report

## Abstract

To the best of the author’s knowledge, studies of mature plasmacytoid dendritic cell proliferation associated with T lymphoblastic lymphoma were extremely rare in the literature. Here, we report a patient who underwent both mature plasmacytoid dendritic cell proliferation and T lymphoblastic lymphoma. With the findings of lymph node biopsy taken from the right cervical and inguinal regions, we identified eye-catching mature plasmacytoid dendritic cells that were considered to be responsible for this lesion at the beginning, until the immunostaining of Ki67 and TDT showed a small group of positive cells hiding in these plasmacytoid dendritic cells. A bone marrow biopsy was also performed on this patient. Microscopically, the hematopoietic tissue was almost completely replaced by lymphoblastoid cells with condensed chromatin, inconspicuous nucleoli and scanty cytoplasm, which were basically the same as those seen in the lymph nodes in morphology. However, there was no sign of plasmacytoid dendritic cells or Langerhans cells in the bone marrow biopsy. With the help of bone marrow biopsy, our final diagnosis of the lymph node was T lymphoblastic lymphoma coexisting with mature plasmacytoid dendritic cell proliferation. Although accumulations of plasmacytoid dendritic cells may occur in some infections or reactive lymphadenopathy, the presence of extensive nodules or infiltration of plasmacytoid dendritic cells strongly reminds the pathologist to carefully evaluate the bone marrow or peripheral blood status of the patient to exclude a hidden myeloid or other neoplasm.

## 1. Introduction

Plasmacytoid dendritic cells (pDCs), first described by Lennert and Remmele in reactive lymphoid hyperplasia in 1958 [1], were initially called ‘plasmacytoid T cells’, ‘T-associated plasma cells’ or ‘plasmacytoid monocytes’. They represent a specialized branch of the dendritic cell family with ‘plasmacytoid’ morphology, and a large capacity for producing type I interferon, which was confirmed by the Liu and Colonna groups until the late 1990s [2,3]. Increased pDCs have been observed in a wide range of inflammatory conditions, including infection, hyaline-vascular Castleman disease and Kikuchi–Fujimoto lymphadenopathy, and sometimes involved in a variety of malignancies and autoimmune diseases [4]. However, the roles pDCs played in the tumor microenvironment were still controversial.

Attributing to the differences in clinical course and pathological features, neoplasms derived from pDCs have been divided into two distinct forms: (1) blastic plasmacytoid dendritic cell neoplasm (BPDCN), a rare but highly aggressive tumor derived from precursors of pDCs, which were characterized by cutaneous involvement and leukemic dissemination; (2) mature plasmacytoid dendritic cell proliferation (MPDCP) associated with myeloid neoplasm, which shows sheets or nodules proliferation of mature plasmacytoid dendritic cells, and has been introduced as a distinct entity in the 5th edition of the WHO Classification of Tumours of Heamatopoietic and Lymphoid Tissues recently [5]. By immunochemistry, both of the entities may express pDC markers, such as CD123, CD303, TCL1, TCF4, etc. However, the former commonly shows strong expression of CD56, while the CD56 expression in the latter is mostly negative, but focal and weak reactivity also can be seen in some cases [6].

According to the definition, MPDCP is invariably associated with a myeloid disorder, predominantly with chronic myelomonocytic leukemia (CMML), but acute myeloid leukemia (AML), myelodysplastic neoplasm (MDS) and myeloproliferative neoplasm (MPN) may also be reported [7]. Whether the MPDCP serves as part of the entire tumor process, or as an independent tumor component remains unclear. Several studies have suggested that MPDCP and their associated myeloid neoplasm cells may be clonally related, but further research is still going on [8,9,10,11]. However, to the best of the author’s knowledge, cases of MPDCP associated with T lymphoblastic lymphoma have been seldom reported in the literature [7,12]. Here, we report a patient who underwent both MPDCP and T lymphoblastic lymphoma.

## 2. Case Presentation

A 65-year-old woman suffered from cough, lymphadenopathy and intermittent high fever for 8 months. The patient was admitted to our hospital after ineffective treatment with anti-inflammatory and anti-tuberculosis therapy at an external medical facility. Her peripheral blood analysis showed leukocytopenia (2.68 × 10^9^/L), normochromic anemia (erythrocyte 2.84 × 10^12^/L, hemoglobin 91 g/L; mean corpuscular volume 95.1 fl), and a normal platelet count (215 × 10^9^/L). Differential counts of white blood cells revealed a relatively high percentage of lymphocytes (72.7%) and a low percentage of neutrophils (21.29%). Meanwhile, her physical examinations showed enlarged lymph nodes over the clavicular, cervical, axillary and inguinal regions, without hepatomegaly or splenomegaly. The PET-CT imaging results show multiple enlarged lymph nodes and increased metabolism in bilateral neck, armpits, retroperitoneum, mesenteric roots, abdominal cavity, bilateral iliac vessels and bilateral inguinal regions.

Lymph node biopsy samples were taken from the right cervical and inguinal regions of this patient. All tissues were fixed in 10% buffered formalin, and then the paraffin-embedded sections were stained with hematoxylin and eosin. Histological examination showed that the normal structure of the lymph nodes was effaced, and the inguinal lymph node was almost entirely replaced by sheets of small- to medium-sized cells with amphophilic cytoplasm, round to ovoid nuclei, finely dispersed chromatin and inconspicuous or small nucleoli (Figure 1A and Figure 2). Immunohistochemistry indicated that the clusters of cells expressed CD123, TCF4, CD4, CD68, CD43, CD31 and LCA, but negative for CD3, CD56, TCL1, CD8, CD20, CD34, S100, CD1a, Langerin, CD61, CD15, CD163, CD117, MPO, ALK and TDT. The proliferation rate of these cells was very low, and only 5% of cells showed nuclear staining for Ki-67. According to the morphology and immunophenotype, these cells were believed to be mature pDCs.

These eye-catching mature plasmacytoid dendritic cells were considered to be responsible for this lesion at the beginning, until the immunostaining of Ki67 and TDT showed a small group of positive cells hiding in these pDCs. Mostly, they were scattered in the paracortex of the lymph nodes, and a few of them formed small clusters. Morphologically, these cells resembled lymphoblasts, characterized by condensed chromatin, inconspicuous nucleoli and scanty cytoplasm (Figure 3A). Immunostaining confirmed that these were T lymphoblasts positive for CD3, CD5, CD7, CD34, LCA and CD31 and negative for CD4, CD8, CD2, MPO and CD117 (Figure 3B). The proliferation rate was very high. However, it was difficult to differentiate reactive lymphoblast proliferation from bona fide lymphoblastic lymphoma in this situation due to the scarcity of these lymphoblasts.

In a few areas of the lymph node, clusters of pDCs mixed with some cells that had abundant, pale cytoplasm, formed a ‘dark–light’ pattern (Figure 1B). The pale cells had distinctively complex folded and grooved nuclei, inconspicuous nucleoli, and finely dispersed chromatin. Binucleated or multinucleated cells may also be found occasionally. The expression of S100, langerin and CD1a revealed that these cells were proliferative Langerhans cells.

The pathological features of cervical lymph nodes were milder than those of inguinal lymph nodes, but the growth pattern and cell composition were quite the same. In conclusion, these lesions are mainly composed of mature pDCs, along with a relatively small amount of Langerhans cells and T lymphoblasts. A bone marrow biopsy was also performed on this patient. Microscopically, the hematopoietic tissue was almost completely replaced by lymphoblastoid cells with condensed chromatin, inconspicuous nucleoli and scanty cytoplasm, which were basically the same as those seen in the lymph nodes in morphology (Figure 4A). However, there was no sign of pDCs or Langerhans cells in the bone marrow biopsy. These lymphoblastoid cells expressed TDT, CD3, CD5, CD7, CD34 and LMO2 but were negative for CD20, CD4, CD117, MPO, CD68, CD123, S100, CD235 and CD61 (Figure 4B). The proliferation rate reached approximately 80%. Flow cytometry showed 85.06% abnormal T lymphoblasts, which had the same immunophenotype of bone marrow biopsy and no sign of pDCs. Next-generation sequencing (NGS) of the pDCs scraped from the inguinal lymph node section revealed a few mutations: DNMT3A (NM_022552): c.G1646T p.Cys549Phe (32.18% VAF), DNMT3A (NM_022552): c.A2321C p.Glu774Ala (28.20% VAF), NF1 (NM_001042492): c.C4600T p.Arg1534Ter (1.61% VAF) and ERCC1 (NM_202001): c.T962C p.Phe321Ser (1.35% VAF). While the NGS results of the T lymphoblasts in the bone marrow biopsy was quite different from pDCs, which demonstrating mutations as follows: CSF3R (NM_000760): c.2188C>T p.Q730X(47.56%,VAF), NOTCH1 (NM_017617): c.4721T>A p.L1574Q (31.85%,VAF) and NOTCH1 (NM_017617): c.5153T>C p.I1718T (4.01%,VAF). Collectively, the diagnosis of bone marrow biopsy was T lymphoblastic lymphoma/lymphoblastic leukemia. With the help of the bone marrow biopsy, our final diagnosis of the lymph node was T lymphoblastic lymphoma coexisting with MPDCP.

After diagnosis, the patient received chemotherapy of Cyclophosphamide + Doxorubicin hydrochloride + Vindesine + Dexamethasone, after two cycles of treatment, the patient developed myelosuppression accompanied by lung infection, and then she was unable to tolerate the side effects of chemotherapy and chose to be discharged, unfortunately, she was lost to follow-up after that.

## 3. Discussion

The origin and characteristics of pDCs have perplexed scientists for decades since they were first described by Lennert and Remmele in 1958. The physiologic, immunologic roles and pathologic states of pDCs have been well studied during the past two decades. They are characterized by a ‘plasmacytoid’ morphology resembling plasma cells which may produce high levels of type I interferons (IFN-I) [13]. pDCs originate in the bone marrow, where they comprise 0.1–0.5% of nucleated cells and then circulate in the peripheral blood as mature cells, which means they remain in a non-proliferative state and survive for only several days [14]. pDCs normally reside in small amounts in lymphoid organs, such as lymph nodes and tonsils, and are seldomly found in bone marrow, spleen, thymic medulla or mucosa-associated lymphoid tissue [15].

Morphologically, pDCs are medium-sized cells with round to ovoid, sometimes slightly elongated nuclei, fine chromatin and moderately abundant cytoplasm that is eosinophilic with hematoxylin/eosin staining and basophilic with Giemsa staining. They are usually situated in the paracortex in lymph nodes, around high endothelial venules as clusters or dispersed cells, while the clusters are scarcely seen in other lymph tissues or bone marrow [16]. PDCs are more readily identified by immunostaining, which is distinguishable by the expression of CD123, CD4, CD68, TCL1 and CD303 (BDCA2). CD2, CD5, CD7 or CD56 may be seen in small proportions of PDCs, usually with only focal and weak reactivity.

In many kinds of inflammatory conditions, such as tumors, autoimmunity and infections, pDCs may become a home to the diseased tissue, usually lymph nodes and skin. In lymph nodes, the increase in pDCs is most commonly seen in three special forms of reactive lymphadenopathy: Kikuchi–Fujimoto lymphadenopathy (histiocytic necrotizing lymphadenitis, Castleman disease (hyaline–vascular subtype) and Kimura disease. In addition, pDCs may infiltrate a variety of malignancies, such as melanoma, squamous cell carcinoma, basal cell carcinoma and breast carcinoma. The properties of pDCs in tumors were controversial. Many studies have suggested that PDCs may enhance an antitumor immune response by using TLR7 and TLR9 agonists [17,18,19]. However, others believe that pDCs may exhibit a pro-tumorigenic or immunoregulatory feature in tumor [20,21].

Owing to the large differences both in clinical course and pathology, neoplasms derived from pDCs have been divided into two distinct forms: mature plasmacytoid dendritic cell proliferation (MPDCP) associated with myeloid neoplasm and blastic plasmacytoid dendritic cell neoplasms (BPDCN), a newly recognized highly aggressive neoplasms, which most likely arise from pDC precursor cells that characteristically express TCF4, TCL1, CD123 and CD4 but also CD56. The former comprises large numbers of mature pDCs with a very low proliferation rate (Ki67 < 10%) and is invariably associated with myeloid neoplasms [7]. In most cases, mature pDCs express TCF4, CD123, CD4, CD68, TCL1 and CD303, but usually negative or only focal/weak positive for CD56. They usually form compact, well-defined nodules in the lymph nodes, bone marrow, spleen or skin, but diffuse infiltration may also be seen in a few cases. The MPDCP with myeloid neoplasm was first described as “plasmacytoid T-cell lymphoma” in 1983 [22]. To date, fewer than 100 cases have been reported as single-case or small-case series. Most of these cases occurred in the bone marrow, followed by the skin, lymph nodes and, more rarely, spleen. Focusing on lymph node lesions, 19 reported cases were found until May 2022 by a literature search [8,10,22,23,24,25,26,27,28,29,30]. Detailed clinical and pathological features from all 20 (the present case and 19 reported cases) patients are provided in Table 1. Elderly patients accounted for the majority of cases (ranging from 6–86 years, mean age 59 years), with the exception of a 6-year-old girl. Tumors were more common in male patients (70%). All patients presented with lymphadenopathy (20/20), followed by hepatomegaly (13/20) and splenomegaly (11/20). Other commonly observed symptoms include weakness, night sweats, weight loss, skin lesions, etc. Most patients (18/20) had myeloid disorders, including chronic myelomonocytic leukemia (6/18), myeloproliferative disorder (6/18) and acute myeloid leukemia (6/18). However, there were two exceptions: Facchetti et al. reported a case of MPDCP coexisting with acute non-B non-T lymphoblastic leukemia, and the present patient showed MPDCP with T lymphoblastic lymphoma/leukemia.

Histologically, pDCs in all the cases were easily identified by their plasmacytoid morphology with medium-sized cells, round to ovoid, sometimes slightly elongated nucleus, fine chromatin and moderately abundant cytoplasm. Lymph node architecture effacement was observed in over 90% of these cases (15/16). Only one case showed clusters of pDCs with a nondestructive growth pattern. In other cases, the pattern of pDC accumulation can be roughly divided into two groups: nodular and diffuse patterns. Usually, pDCs formed compact and well-demarcated nodules (10/19), and a diffuse pattern of infiltration was observed in 33.3% (6/19) of these cases. In addition, a mixed pattern of both nodular and diffuse growth was revealed in three patients. Its site of predilection was the paracortical area, while the cortex and medulla area can also be involved in a few cases. Of interest, although the diagnosis of myeloid neoplasms or lymphoblastic lymphoma/leukemia was confirmed by bone marrow biopsy, more than 80% (14/17) of cases showed lymph node involvement of myeloid or lymphoblastic tumors. These myeloid or lymphoblastic tumor cells usually accounted for a small number, were dispersed or formed small clusters along the trabecular septa and medulla, sometimes within the paracortical area or capsule. In a few rare cases, large numbers of myeloid or lymphoblastic tumor cell infiltration can also be found. The expression of immunohistochemical markers was similar to that in normal plasmacytoid dendritic cells, and the most useful markers were TCF4, CD4, CD68, CD123 and Ki67 (low proliferation rate). B cell or T cell markers were often negative, with occasional aberrant expressions of CD2, CD5, CD7 and CD10. It is worth noting that, unlike its counterpart BPDCN, CD56 was negative in the majority of the cases, with only focal and weak reactivity observed in a few cases. Although the pDCs may be superior in number, the prognosis relies on the patient’s underlying myeloid neoplasm rather than the expansion of pDCs. All patients underwent chemotherapy, but the prognosis was usually dismal. The median follow-up duration was 26 months, ranging from 17 days to 7 years, and 14 (82%) of 17 patients had died. Thus, the five-year OS rate was 11.7%.

The nature of nodal accumulations of PDCs in patients with myeloid neoplasms has perplexed the pathologist and clinician for quite a long time. Lymph node parenchyma effacement can be seen in almost all patients, and aberrant expression of CD2, CD5, CD7, CD10 or CD56 in some cases favors a neoplastic origin of these cells [7]. Recently, via fluorescence in situ hybridization, an increasing number of studies have suggested that neoplastic mature plasmacytoid dendritic cells and their associated myeloid neoplasm cells share similar and clonal chromosomal abnormalities. A case from Vermi et al. demonstrated monosomy 7 in both myeloid leukemia and PDC nodules [8]. Monosomy 7 was also observed in two kinds of cellular components in one case reported by Chen et al., and they reported another case showing a loss of 20q12 in both populations [11]. One case of cutaneous pDCs associated with CMML also presented with trisomy 13 in both leukemia cells and neoplastic pDCs [9]. In addition, with the development of massive parallel sequencing (NGS), Bodmer et al. revealed a common PTPN11 gene mutation shared by the MDS and pDC populations [10]. Despite this genetic evidence, hints provided by the clinical course of pDCs paved the way for the coming truth. Follow-up of a patient by Harris et al. showed that the non-biopsied lymph node did not enlarge but regressed after being given prednisone and busulphan [29]. In addition, Dargent et al. revealed an interesting phenomenon of simultaneous regression of cutaneous pDC accumulation and reduction of peripheral leukemia cells after therapy aimed at leukemia [9]. All of these findings led some researchers to consider pDC proliferation as part of the entire tumor process, other than an independent tumor component, and both components may share a common myeloid precursor-cell origin [31].

Apart from the present case, only two cases of MPDCP with lymphoblastic leukemia were reported [12,27]. However, due to the nondestructive growth pattern of the pDC part, the author considered the pDC part as the reactive component of the tumor in one case [7,12]. Different from the former case, the growth pattern and pathogenic mutation of the pDCs in the present case demonstrated their neoplastic nature. Another case of MPDCP associated with T-lymphoblastic leukemia was recently reported by Oscar Sliva et al. in bone marrow, while the limitation of this report lies in the unsorting NGS of pDCs and T lymphoblasts components.

By definition, MPDCP with myeloid neoplasm is a tumor characterized by mature plasmacytoid dendritic cell proliferation invariably associated with a myeloid neoplasm [7]. While in the present case, the tumor was mainly composed of neoplastic pDC proliferation, along with a relatively small amount of T lymphoblast lymphoma cells and Langerhans cells, the question of whether to put the case into MPDCP with myeloid neoplasm remains to be answered. Of interest, Langerhans cells mixed with pDCs seen in the present case were also observed in some reports of MPDCP with myeloid neoplasm [30,32,33]. Therefore, initially, we hypothesized that there may be a hematopoietic stem cell at the very beginning of those MPDCP-associated tumors. It may have multilineage potential to give rise to neoplastic cells of different lineages under different microenvironments. However, the mutations detected by NSG were completely different in pDCs and T lymphoblasts components, which goes against the initial assumption, MPDCP and T lymphoblastic lymphoma may be genetically unrelated to each other.

## 4. Conclusions

Thorough research about mutational clonality between MPDCP and lymphoid neoplasms is still needed to illustrate the uncommon condition. Although studies focused on MPDCP have lasted more than 30 years, the cell lineage is still not definitely understood. Whether MPDCP can occur in both lymphoid and myeloid tumors also remains to be elucidated in the future.

The significance of our case lies in two aspects: (1) to report a rare case of mature pDC proliferation coexisting with T lymphoblastic lymphoma/leukemia and reveal different mutations between mature pDC proliferation and T lymphoblastic lymphoma/leukemia; (2) although accumulations of pDCs may occur in some infections or reactive lymphadenopathy, the presence of extensive nodules or infiltration of pDCs strongly reminds the pathologist to carefully evaluate the bone marrow or peripheral blood status of the patient to exclude a hidden myeloid or other neoplasms.

## Figures and Tables

**Figure 1 diagnostics-13-03248-f001:**
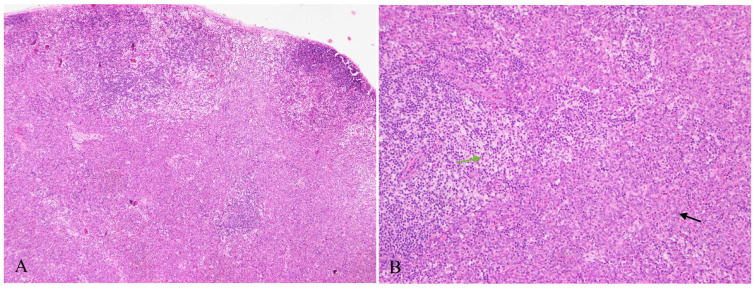
(**A**) Normal structure of lymph nodes was effaced by sheets of tumor cells with amphophilic cytoplasm (HE ×40). (**B**) Clusters of pDCs (black arrow) mixed with Langerhans cells (green arrow) characterized by abundant and pale cytoplasm, forming a ‘dark–light’ pattern (HE ×100).

**Figure 2 diagnostics-13-03248-f002:**
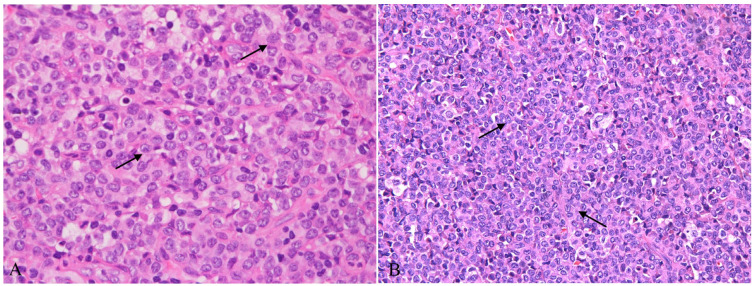
(**A**,**B**) Mature plasmacytoid dendritic cells with amphophilic cytoplasm, round to ovoid nuclei, finely dispersed chromatin and inconspicuous or small nucleoli (HE ×400, black arrow).

**Figure 3 diagnostics-13-03248-f003:**
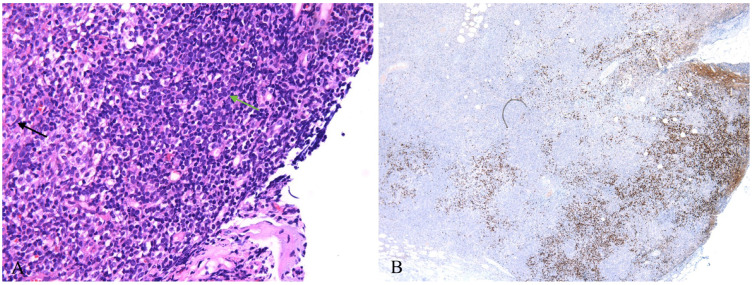
(**A**) Lymphoblasts (green arrow) hiding in mature plasmacytoid dendritic cells (black arrow), featuring condensed chromatin, inconspicuous nucleoli and scanty cytoplasm (HE ×400); (**B**) Scattered lymphoblasts highlighted by TDT stain (IHC ×40).

**Figure 4 diagnostics-13-03248-f004:**
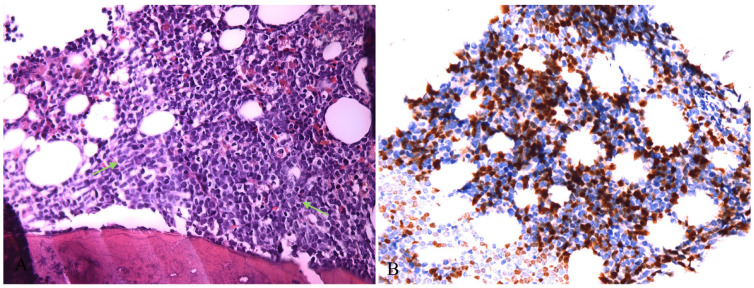
(**A**) Hematopoietic tissue in bone marrow was replaced by lymphoblasts (HE ×400,green arrow); (**B**) Tumor cells showed nuclear expression of TDT (IHC ×400).

**Table 1 diagnostics-13-03248-t001:** **Overview of documented cases of MPDCP coexisting with myeloid/lymphoid tumors in lymph nodes, as reviewed from literature until May 2022.**

Report	Case	Gender	Age	Pattern	Destruction of Normal Structure	Location	Nodular Myeloid or Lymphoid Tumor	Associated Tumor	Presentation	Follow-Up Time	Outcome
Muller et al., 1983 [22]	1	M	65	NM	NM	NM	NM	Acute myelomonocytic leukemia	fatigue, weight loss, L, H, S	7 M	Dead
Grizzle et al., 1985 [23]	2	M	86	D	YES	NM	NO	Chronic myelogenous leukemia	weight loss, L, H, S	3 W	Dead
Beiske et al., 1986 [24]	3	M	74	N	NM	P	YES	Acute myelomonocytic leukemia	Weight loss, night sweat, L	6 M	Dead
Thomas et al., 1991 [25]	4	F	6	D	YES	P	NO	Atypical myeloproliferative disorder	L, H, S	NM	NM
Koo et al., 1990 [26]	5	F	58	D	YES	P, M	YES	Myeloproliferative disorder	anemia, night sweat, L, H, S	28 M	Dead
Facchetti et al., 1990 [27]	6	M	75	N D	YES	P, C	YES	Chronic myelomonocytic leukemia	Weight loss, L, H, S	16 M	Dead
	7	M	66	N	NO	P	YES	Acute non-B non-T lymphoblastic leukemia	dyspnea fever, L, H, S	20 D	Dead
Baddoura et al., 1992 [28]	8	M	58	D	YES	P, C, M	YES	Chronic myeloproliferative disorder	Fatigue fever weight loss, L, H, S	Lost	Lost
	9	M	73	N	NM	NM	NM	Acute monocytic leukemia	weight loss, fatigue, L	Lost	Lost
Harris et al., 1991 [29]	10	F	54	N	YES	P	YES	Chronic myelomonocytic leukemia	fatigue, weight loss, L, S	84 M	Dead
Vermi et al., 2004 [8]	11	M	24	N D	YES	NM	YES	Chronic myelomonocytic leukemia	L, H, S	8 M	Dead
	12	M	50	N	YES	NM	YES	Acute myelomonocytic leukemia	L, H, S	11 M	Alive
	13	M	58	N	YES	NM	YES	Chronic myelomonocytic leukemia	L, H	84 M	Dead
	14	F	63	N	YES	NM	YES	Unclassifiable chronic myeloproliferative disorder;	L, H	15 M	Dead
	15	M	80	D	YES	NM	NO	Unclassifiable myeloproliferative/myelodysplastic disorder	L, H, S	43 M	Dead
	16	F	62	N	YES	NM	YES	Acute monocytic leukemia	L	15 M	Dead
	17	M	52	N	YES	NM	YES	Chronic myelomonocytic leukemia	L, H	13 M	Alive
Song et al., 2012 [30]	18	M	55	N D	YES	NM	YES	Acute myeloid leukemia	weigh loss, L	17 D	Dead
Bodmer et al., 2017 [10]	19	M	65	N	NM	P	NM	Myelodysplastic syndromes	L	28 M	Dead
present	20	F	65	D	YES	P, C, M	YES	T lymphoblastic lymphoma/leukemia	L	3 M	Alive

Abbreviations: NM, not mention; N, nodule; D, diffuse; P, paracortical area; C, cortex; M, medulla area; L, lymphadenopathy; H, hepatomegaly; S, splenomegaly.

## Data Availability

Data availability is not applicable to this article as no new data were created or analyzed in this study.

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
