# Peer review of "T Lymphoblastic Lymphoma Hiding in Mature Plasmacytoid Dendritic Cell Proliferation: A Case Report and Literature Review"

_diagnostics, 2023, doi:10.3390/diagnostics13203248_

Round 1

Reviewer 1 Report

This article is particularly of interest in the field of MPDCP, and relatively easy to read. Just some few notes :

Line 30 : the term « plasmacytoid dendritic cell » is generally abbreviated « pDC » rather than « PDC », but this choice is left to the discretion of the authors.

Line 42 : the choice of the author was to prefer MPDMN to MPDCP with MN. The more recent term in the WHO (2017 and 2022) is mature plasmacytoid dendritic cell proliferation associated (MPDCP) with myeloid neoplasm, to my knowledge, but it is maybe a new term adopted in 2023 ?

Line 47 : please add the references 15 and 29 ; similarly to line 236

Line 61-62 : « A bone marrow aspiration examination demonstrated that bone marrow was invaded by the tumor cells. » => further exploration are presented below line 110 => maybe put this sentence just befor line 110

Line 62-67 => after line 127 would be more appropriate

Line 74 : the notion of clusters may be added to the text, considering that this is an usual aspect of MPDCP in CMML, and that this term is mentioned in Figure 1.

Line 75 : it may be useful to reorder the markers so that to emphasize the one that are the more discriminant and expressed on pDCs (CD56 may also be emphasized, for BPDCN differential diagnosis) : « CD123, TCF4, CD4, CD68, CD43, CD31 and LCA but negative for CD3, CD56, TCL1, CD8, CD20, CD34, S100, CD1a,... »

Line 82 : « with some cells » should be replaced by « with Langerhans cells » to better understand the purpose ? 

Line 89 : « hiding in the pDC infiltrate » seems more appropriate, these cells are not « in » pDCs but « with » pDCs

 Line 95 : a percentage would be welcome, for example : « probably over xx% (70/80/90% ?), However, it was difficult to quantify because hard to differentiate …»

Line 119 : « no sign of pDC » using what gating strategy ? CD123/CD4 ? CD123/HLA-DR ? Other ?

Ligne 119 : please provide the way to obtain pDCs : which biopsy ? cell sorting ? Idem line 122 for T lymphoblasts.

Line 120 : NGS result : please provide the NM that was used of each gene, and the c. coordinates of the mutations.

Line 137 : maybe to be tempered, antigen-presenting properties are lower thant cDCs in canonical pDCs. APC properties would be attributable to tDCs/AS-DCs. We actually not really know if canonical pDCs are real APC, I think ?

Line 159 : conflicting properties of pDCs in tumors : antitumor/protumor ? Depends on the infiltrate ? (in CML : Inselmann et al., 2018), other data of pDCs associated with progression (tumor-associated pDCs) 

Line 164 : « CD123, CD4 and CD56 » => CD56 should be emphasized, because rarely expressed in physiological pDCs, maybe « CD123, CD4 but also CD56 … (…in contrast to a majority pDCs for exemple).

 Line 167 : « CD123, CD4 ; CD68, TCL1 » => please add TCF4, considering that it is one of your more contributive staining in your panel.

 Line 181-182 and in the Table : « acute myelomonocytic leukemia (3/18), acute monocytic leukemia 181 (2/18), and acute myeloid leukemia » may be merged in AML, to be easier to read.

Similarly, in the table, « Atypical myeloproliferative disorder, Chronic myeloproliferative disorder, Unclassifiable chronic myeloproliferative disorder » should be merged in « myeloproliferative disorder »

 Line 205 : the more specific markers for pDCs (CD303, TCF4, TCL1, BCL11A…) are also the most useful markers. Please add some of them.

 Line 227 : other major NGS data with mutational status on blasts and on pDCs, especially concerning RUNX1 mutation that deserve to be highlighted (in this lymphoid report there is no RUNX1 mutation => specific to MPDCP associated with MN ?) :

 - Xiao, W.; Chan, A.; Waarts, M.R.; Mishra, T.; Liu, Y.; Cai, S.F.; Yao, J.; Gao, Q.; Bowman, R.L.; Koche, R.P.; et al. Plasmacytoid Dendritic Cell Expansion Defines a Distinct Subset of RUNX1-Mutated Acute Myeloid Leukemia. Blood 2021, 137, 1377–1391. https://doi.org/10.1182/blood.2020007897.

- Zalmaï, L.; Viailly, P.-J.; Biichle, S.; Cheok, M.; Soret, L.; Angelot-Delettre, F.; Petrella, T.; Collonge-Rame, M.-A.; Seilles, E.; Geffroy, S.; et al. Plasmacytoid Dendritic Cells Proliferation Associated with Acute Myeloid Leukemia: Phenotype Profile and Mutation Landscape. Haematologica 2021, 106, 3056–3066. https://doi.org/10.3324/haematol.2020.253740.

Two points I noticed concerning English :

Line 238 : I do not understand this sentence : « Different from the former case, the grow pattern and pathogenic mutation of demonstrated the neoplastic origin of these PDCs. »  Missing word after « mutation of »/or delete « of » ?

Line 256 « still needed » instead of « still need » I think, but I am not an English-speaking

Author Response

On behalf of my co-authors, we are very grateful to you for giving us an opportunity to revise our manuscript. We appreciate you very much for your positive and constructive comments and suggestions on our manuscript entitled “T lymphoblastic lymphoma hiding in mature plasmacytoid dendritic cell proliferation: a case report and literature review”(ID: diagnostics-2623671). We have studied reviewers' comments carefully and tried our best to revise our manuscript according to the comments. The following are the responses and revisions I have made in response to the reviewers' questions and suggestions on an item-by-item basis. Thanks again to the hard work of the editor and reviewer!

Response to the comments of Reviewer #1:

Line 30: the term « plasmacytoid dendritic cell » is generally abbreviated « pDC » rather than « PDC », but this choice is left to the discretion of the authors.

Thank you for your meticulous observation. Due to our oversight, we made this error. We have made the correction in the manuscript. Please see the attachment.

Line 42: the choice of the author was to prefer MPDMN to MPDCP with MN. The more recent term in the WHO (2017 and 2022) is mature plasmacytoid dendritic cell proliferation associated (MPDCP) with myeloid neoplasm, to my knowledge, but it is maybe a new term adopted in 2023?

After consulting the literature and books, we have determined that your suggestion is more appropriate. We have made the necessary changes in the manuscript. Please see the attachment.

Line 47: please add the references 15 and 29; similarly to line 236.

Thank you for pointing this out. We have added the references in the manuscript. Please see the attachment.

Line 61-62: « A bone marrow aspiration examination demonstrated that bone marrow was invaded by the tumor cells. » => further exploration are presented below line 110 => maybe put this sentence just before line 110.

After careful consideration, we believe that your suggestion provides a more logical flow to the manuscript. We have made the necessary changes. Please see the attachment.

Line 62-67 => after line 127 would be more appropriate.

After careful consideration, we believe that your suggestion provides a more logical flow to the manuscript. We have made the necessary changes. Please see the attachment.

Line 74: the notion of clusters may be added to the text, considering that this is a usual aspect of MPDCP in CMML, and that this term is mentioned in Figure 1.

Thank you for your reminder. We have added the notion of clusters in the manuscript. Please see the attachment.

Line 75: it may be useful to reorder the markers so that to emphasize the ones that are more discriminant and expressed on pDCs (CD56 may also be emphasized, for BPDCN differential diagnosis): « CD123, TCF4, CD4, CD68, CD43, CD31 and LCA but negative for CD3, CD56, TCL1, CD8, CD20, CD34, S100, CD1a,... »

We completely agree with your perspective and have made the necessary changes in the manuscript. Please see the attachment.

Line 82: « with some cells » should be replaced by « with Langerhans cells » to better understand the purpose?

We completely agree with your perspective and have made the necessary changes in the manuscript. Please see the attachment.

Line 89: « hiding in the pDC infiltrate » seems more appropriate, these cells are not « in » pDCs but « with » pDCs.

We completely agree with your perspective and have made the necessary changes in the manuscript. Please see the attachment.

Line 95: a percentage would be welcome, for example: « probably over xx% (70/80/90% ?), However, it was difficult to quantify because hard to differentiate …»

We apologize, but it's challenging to provide a specific percentage as it's difficult to differentiate.

Line 119: « no sign of pDC » using what gating strategy? CD123/CD4? CD123/HLA-DR? Other?

We apologize for the oversight. The observation of "no sign of pDC" is based on our histological and immunohistochemical indicators. The following are the flow cytometry results of the patient's bone marrow: 85.06% of all nucleated cells expressed CD7, CD34, cCD3, CD38, CD33, CD5, CD99 ,TdT, and did not express CD117, CD19, MPO,cCD79a, CD22,CD10,CD56, CD2,CDla, CD5, CD4, CD8, CD3,CD13, HLA-DR, CD138. The phenotype indicates abnormal T immature cells; Conclusion: It is likely ALL-T (most likely at the pro-T stage).

Line 119: please provide the way to obtain pDCs: which biopsy? cell sorting? Idem line 122 for T lymphoblasts.

Line 119: Thank you for pointing this out. The pDCs were obtained from the scraped section of the inguinal lymph node, while the T lymphoblasts were sourced from the bone marrow biopsy. We have added the method of obtaining pDCs in the manuscript. Please see the attachment.

Line 120: NGS result: please provide the NM that was used for each gene, and the c. coordinates of the mutations.

Thank you for your reminder. We have added the NM used for each gene and the c. coordinates of the mutations in the manuscript. Please see the attachment.

Line 137: maybe to be tempered, antigen-presenting properties are lower than cDCs in canonical pDCs. APC properties would be attributable to tDCs/AS-DCs. We actually not really know if canonical pDCs are real APC, I think?

Thank you for pointing out this issue. There is currently no consensus on whether pDCs can act as genuine APCs. We have removed the related content from the article.

Line 159: conflicting properties of pDCs in tumors: antitumor/protumor? Depends on the infiltrate? (in CML: Inselmann et al., 2018), other data of pDCs associated with progression (tumor-associated pDCs).

We have reviewed the relevant literature and added its content. Please see the attachment.

Line 164: « CD123, CD4 and CD56 » => CD56 should be emphasized, because rarely expressed in physiological pDCs, maybe « CD123, CD4 but also CD56 … (…in contrast to a majority pDCs for example).

We completely agree with your perspective and have made the necessary changes in the manuscript. Please see the attachment.

Line 167: « CD123, CD4; CD68, TCL1 » => please add TCF4, considering that it is one of your more contributive staining in your panel.

We completely agree with your perspective and have made the necessary changes in the manuscript. Please see the attachment.

Line 181-182 and in the Table: « acute myelomonocytic leukemia (3/18), acute monocytic leukemia 181 (2/18), and acute myeloid leukemia » may be merged in AML, to be easier to read. Similarly, in the table, « Atypical myeloproliferative disorder, Chronic myeloproliferative disorder, Unclassifiable chronic myeloproliferative disorder » should be merged in « myeloproliferative disorder ».

Thank you for your suggestion. We have made the necessary changes in the manuscript. Please see the attachment.

Line 205: the more specific markers for pDCs (CD303, TCF4, TCL1, BCL11A…) are also the most useful markers. Please add some of them.

Thank you for your reminder. We have added the more specific markers for pDCs in the manuscript. Please see the attachment.

Line 227: os lymphoid report there is no RUNX1 mutation => specific to MPDCP associated with MN?):

Neither the plasmacytoid dendritic cell component nor the T-lymphoblastic lymphoma/leukemia component showed any RUNX1 mutation.

Line 238: I do not understand this sentence: « Different from the former case, the grow pattern and pathogenic mutation of demonstrated the neoplastic origin of these PDCs. » Missing word after « mutation of »/or delete « of »?

Thank you for pointing this out. What we really want to express is “Different from the former case, the grow pattern and pathogenic mutation of the pDCs in the present case demonstrated their neoplastic nature.”We have made the necessary corrections in the manuscript. Please see the attachment.

Line 256 « still needed » instead of « still need » I think, but I am not an English-speaking.

Thank you for pointing this out. We have made the necessary corrections in the manuscript. Please see the attachment.

I hope this meets your requirements. If there are any further changes or edits needed, please let me know.

Reviewer 2 Report

Dear authors,

Plasmacytoid dendritic cells are frequently associated with myeloid malignancies. One of these is blastic plasmacytoid dendritic cell neoplasm and another, chronic myeloid leukemia. The association between lymphoblastic leukemia and plasmacytoid dendritic cells has been little explored. The diagnosis was established based on a comprehensive review of morphology, immunophenotype, and clinical implications by the authors. Prominent proliferation of plasmacytoid dendritic cells may be associated with lymphoid neoplasms and may present with blastic morphology and immunophenotype. The underlying mechanism of the coexistence of these two blast populations is still unknown in the literature, in the context of rare exhibitions of this type of cases. Additional genetic profiling may be required to indicate progression of tumor stem cells to lymphoid, myeloid, or dendritic cell lineages. From this perspective, of the rarity of the case presented, the work is valuable.

From my point of view, a more exhaustive introduction would be necessary. It also means better clarity of histology images. Exemplifying the changes using colored arrows should make the image more accessible to a wider medical public, not only to oncologists and histopathologists. Separation of conclusions is required. The article is documented, it presents a rare pathology and with a little effort on the part of the authors it could be proposed for publication.

Author Response

On behalf of my co-authors, we are very grateful to you forgiving us an opportunity to revise our manuscript. We appreciate you very much for your positive and constructive comments and suggestions on our manuscript entitled “T lymphoblastic lymphoma hiding in mature plasmacytoid dendritic cell proliferation: a case report and literature review”(ID: diagnostics-2623671). We have studied reviewers' comments carefully and tried our best to revise our manuscript according to the comments. The following are the responses and revisions I have made in response to the reviewers' questions and suggestions. Thanks again to the hard work of the editor and reviewer!

Response to the comments of Reviewer #2:

Thank you for your suggestions. After reviewing our article, we indeed found that the introduction section needs to be more detailed, and the conclusion section should be separated. Therefore, we have made the necessary modifications in the manuscript. Additionally, based on your feedback, we have added arrows to the images in the article, making it easier and more effective for readers to understand.

I hope this meets your requirements. If there are any further changes or edits needed, please let me know.

Reviewer 3 Report

The article has a typical layout, consistent with the journal's guidelines. The authors touched on a very important topic. The paper presents T lymphoma, both on the basis of a clinical case and available literature. The authors describe basic and specialized research. The discussion concludes with two important conclusions: to represent rare cases of proliferation of mature PDC coexisting with lymphoma/T-lymphoblastic leukemia, and to remind the pathologist to carefully evaluate the patient's bone marrow or peripheral blood status to exclude occult myeloid or other malignancy.

Author Response

On behalf of my co-authors, we are very grateful to you for giving us an opportunity to revise our manuscript. We appreciate you very much for your positive and constructive comments and suggestions on our manuscript entitled “T lymphoblastic lymphoma hiding in mature plasmacytoid dendritic cell proliferation: a case report and literature review”(ID: diagnostics-2623671). We have studied reviewers' comments carefully and tried our best to revise our manuscript according to the comments. The following are the responses and revisions I have made in response to the reviewers' questions and suggestions. Thanks again to the hard work of the editor and reviewer!

Response to the comments of Reviewer #3:

    Thank you for your thoughtful and positive feedback on our manuscript. We appreciate your recognition of the importance of the topic we've addressed and the manner in which we've presented both the clinical case and the available literature. We concur with the significance of the conclusions drawn and are grateful for your emphasis on the need for pathologists to be thorough in their evaluations. We believe that sharing such rare cases and insights will contribute to the broader understanding in the field.

If there are any further changes or edits needed, please let me know.

Round 2

Reviewer 2 Report

Thank you for the respons and the modifications in your work. I still think that the introduction can be enlarged. In rest I don't have more questions.

Author Response

Dear Editor,

Thank you for your feedback on our revised manuscript.

We have made efforts to address the concerns you previously raised, including enhancing the introduction. However, we understand and respect your perspective that the introduction can still be expanded further. We believe the current version(Please see the attachment) provides a more comprehensive overview, but we are open to specific suggestions or areas you believe would benefit from further elaboration.Your guidance is invaluable to us, and we aim to ensure the manuscript meets the highest standards of the journal. If there are particular points or topics you'd like us to delve deeper into within the introduction, please let us know. We are more than willing to make any further adjustments as needed.

Thank you once again for your patience and understanding.

Warm regards
